# Critical Evaluation of the Role of Transcription Factor RAR-Orphan Receptor-γt in the Development of Chronic Inflammatory Dermatological Diseases: A Promising Therapeutic Target

**DOI:** 10.3390/biom15111543

**Published:** 2025-11-02

**Authors:** Anik Pramanik, Pallabi Mondal, Sankar Bhattacharyya

**Affiliations:** Department of Zoology, Academic Building II, Sidho Kanho Birsha University, PO: Sainik School, Ranchi Road, Purulia 723104, West Bengal, India; witchera24@gmail.com (A.P.); plbimondal@gmail.com (P.M.)

**Keywords:** chronic inflammation, dermatitis, orphan receptors, IL-17, psoriasis, RORγ, RORγt, skin disorder, Th17

## Abstract

Nuclear receptors (NRs) are transcription factors regulated by ligands that direct metabolism, development, and immunity. The NR superfamily constitutes a principal category of pharmacological targets for human ailments. Retinoic acid receptor-related orphan receptors (RORs) α, β, and γ are part of the nuclear receptor superfamily. They are nevertheless classified as “orphan” receptors due to the contentious nature of identifying their endogenous ligands. RORγ nuclear receptor protein further consists of two isoforms, namely RORγ1 and RORγ2 or RORγt. RORγt is largely found in immune cells and has been primarily associated with chronic inflammatory conditions. The expression of STAT3 is a major driver of Th17 differentiation and induces RORγt expression through the JAK-STAT pathway. Type 3 innate lymphoid cells (ILC3s), Th17 cells, and γδT cells express RORγt, the master transcription regulator for the pro-inflammatory cytokine interleukin IL-17. In chronic inflammatory skin disorders, a significant increase in IL-17 has been observed, which plays a key role in both immune cell recruitment to the site of inflammation and the propagation of tissue damage. In this review, we will discuss how RORγt regulates IL-17-driven inflammation and explore potential strategies to target the RORγt-IL-17 axis as a viable therapeutic intervention in chronic inflammatory skin disorders.

## 1. Introduction

The retinoic acid-related orphan receptor (ROR) subfamily belongs to the Nuclear Receptor (NR) 1 subfamily. The NR1 subfamily comprises RORα (RORA), RORβ (RORB), and RORγ (RORC) [1]. The functional activity of RORs is regulated through the generation of different isoforms, produced via alternate splicing or alternative promoter utilization. In both mice and humans, the RORc gene possesses two distinct isoforms: RORγ1 (RORγ) and RORγ2 (RORγt). Initially, RORs were referred to as orphan receptors due to the absence of any known physiologically relevant ligands. However, putative ligands for those receptors have been found, including steroids, cardiac glycosides, terpenoids, and polyketides [2]. Upon ligand binding to the ligand-binding domain (LBD) of RORs, conformational alterations in the receptor occur, facilitating the recruitment of co-regulatory proteins, which eventually enhances the transcriptional activity of RORs [3,4,5].

On one hand, ROR transcription factors attach to certain DNA regions termed ROR response elements (ROREs) and regulate the expression through directly binding to the target genes; on the other hand, they interact with co-regulatory proteins (SRC1, SRC2, SRC3, p300) and transcriptional co-activator TAZ with PDZ-binding motif to influence the target genes [6,7,8,9]. The RORγ, particularly its isoform RORγt, is a crucial transcription factor that significantly impacts the development, differentiation, and function of various immune cell types, encompassing both innate and adaptive immunity. Its pleiotropic functions are essential for not only coordinating immunological responses but also for preserving immune homeostasis, and it plays a crucial function in lymphoid organogenesis and the formation of various immune cells, including but not limited to Th17 cells, regulatory T cells, γδ T17 cells, iNKT cells, and some myeloid cells.

RORγt is essential for the differentiation of pro-inflammatory Th17 cells [10], as it directly regulates the expression of genes essential for Th17 cell functionality, including *IL17A* and *IL17F* [11]. RORγt attaches to particular DNA sequences, termed ROR response elements, located inside the promoter regions of target genes, enhancing their transcription [12]. The dysregulation of the RORγt/IL17 axis is fundamental to the pathogenesis of various autoimmune and inflammatory disorders, including multiple sclerosis [13], rheumatoid arthritis [14], inflammatory bowel disease [15], and respiratory conditions such as asthma [16] and chronic obstructive pulmonary disease [17]. IL-17, predominantly generated by T helper 17 cells but also by γδ T cells and innate lymphoid cells, regulates inflammatory responses by not just targeting keratinocytes and endothelial cells but also various immune cells in the skin as well [18]. The pivotal role of IL-17 in chronic inflammatory dermatological conditions, especially psoriasis, has resulted in the formulation of highly promising treatments that specifically target IL-17 or its receptor. Interventions including inverse agonists, antagonists and natural products have exhibited significant clinical efficacy in ameliorating the pathological characteristics of conditions such as psoriasis, underscoring IL-17 as a vital therapeutic target [19,20,21].

This review summarizes current studies on the role of RORγt in the etiology of several chronic inflammatory skin illnesses and highlights the advancements as well as the challenges in therapeutics that particularly target the RORγt pathway.

## 2. ROR-Gamma: Isoforms, Upstream Ligands and RORγ Activation

The Nuclear Receptor (NR) 1 subfamily includes the retinoic acid-related orphan receptors (RORs). ROR isoforms generated through alternative splicing or alternative promoter usage are readily expressed in a plethora of tissues. Among the isoforms RORα (*RORA*), RORβ (*RORB*), and RORγ (*RORC*) (Figure 1), RORγt is predominantly expressed in immune cells [1]. In both mice (located on chromosome 3, specifically on the 3 F2.1 band) and humans (located on the long arm (q) of chromosome 1, specifically on the 1q21.3), the *RORC* gene has two different isoforms that include RORγ1 (RORγ) and RORγ2 (RORγt) [22] and contains 11 exons in both humans (*RORC*) and mice (*Rorc*). Both isoforms share the last nine exons (exons 3–11). Development of the two distinct isoforms depends only on the differential transcription of exon 1 and exon 2. As a result, the mRNA of RORγ1 is 100 nucleotides larger than the mRNA of RORγt [23]. These isoforms vary only in the N-terminal domain and exhibit different expression patterns and functions. For example, expression of RORγ1 is mostly observed in peripheral tissues like the kidney, liver, muscle, lung, heart, and brain, where it contributes to the maintenance of circadian rhythm and metabolic regulation. Whereas RORγt is mainly expressed in lymphoid tissues, especially in the lymphoid immune cells like Th17 cells, type 3 innate lymphoid cells and double-positive thymocytes, it contributes to immune regulation [24,25,26,27,28]. In 2012, Rauen et al. identified another isoform of RORγ known as RORγt-Δ, missing the hinge region encoding exons 5–8 in humans [29]. RORγt maintains the survival and cell death of CD4 and CD8 double-positive thymocytes. It is required for the development and differentiation of ILC3 and Th17 cells, as well as in the production of IL-17, and is involved in the lymphoid tissue inducer (LTi) cell generation, which contributes to the formation of Peyer’s patches and lymph nodes [30]. RORγt is also expressed in other immune cells like FOXP3+ Treg cells, γσ T cells, invariant NK cells, etc. It has been reported that, in the gut microbiome, FOXP3^+^RORγt^+^ Treg cells are not dependent on RORγt to produce IL-17; rather, the gut microbiome helps to induce them. So, the effect of RORγt is not similar in every cell; it is context-specific [31]. Further, it has been suggested that its expression and function vary within different immune cell subsets and have several roles in the maintenance of immunologic homeostasis and the development of autoimmune diseases.

Generally, most of the nuclear receptors bind their particular ligands in the cytosol, then are transported into the nucleus either in the monomerized or dimerized form. Like other NRs, RORγ also has a DNA-binding domain (DBD) and a carboxy-terminal ligand-binding domain (LBD). The activating domain is responsible for the binding of the co-activator or corepressor. The ligand-binding domain is not only responsible for ligand binding but also plays a significant role in recruiting the regulatory proteins. The DNA-binding domain allows the ligand–receptor–cofactor complex to link with specific response elements. Another flexible and non-structured domain called the hinge region connects the DBD to the LBD. In this hinge region, most of the post-translational modifications occur and are mainly responsible for the transportation of the nuclear receptors in the presence of NLS (nuclear localization signal). This NLS region is generally recognized by importin α [32,33].

As sterols are one of the most important ligands of RORγ, alterations in sterol metabolism can change the transcriptional activities of ROR, which directly influence both normal and disease conditions. RORγ regulates several important metabolic pathways, including gluconeogenesis, lipid metabolism and cholesterol metabolism pathways. It also reported that disruption in the cholesterol biosynthesis pathway can hamper the RORγ-dependent Th17 cell differentiation [34]. Transcriptomic analysis of psoriatic skin revealed that there is a connection between the genes required for cholesterol biosynthesis and the level of IL-17 [35]. Intriguingly, during the differentiation of Th17 cells, the expression of various cholesterol biosynthesis genes like *FDFT1*, LSS, and *DHCR24* was upregulated, while the genes required for cholesterol metabolism (*CYP7A1*, *CYP27A1*) and cholesterol efflux [*ABCA1*, *ABCG1*] showed downregulated expression [36]. Increased gene expression of cholesterol biosynthetic pathways results in an increase in RORγt activity, which ultimately promotes Th17 cell differentiation and IL-17 production, inducing autoimmune and inflammatory diseases.

Expression of RORγt can also be enhanced by IL-6 and TGF-β, which drives the differentiation of Th17 [37]. However, IL-23 is not necessarily required for the Th17 differentiation, but it helps to stabilize the differentiated Th17 cells by inducing the expression of RORγt via RUNX1 [38]. Ultimately, RORγt binds to the ROR response elements on the DNA of *IL17A*, *IL17F*, and *IL23R* and stimulates their transcription [39].

## 3. Downstream Genes: Promoter Regions RORγ Binds to and Response Elements It Activates/Regulates

The nuclear receptors contribute to the regulation of several gene expressions, which influence cellular growth, differentiation and cell death [40]. Recently, it has been found that the superfamily of nuclear receptors majorly contributes to the endocrine physiology and is involved in various pathways that mainly regulate the development and physiology across species [41]. Alteration of these pathways leads to several diseases like diabetes, cancer and cardiovascular diseases.

The ROR transcription factors bind to the specific region of DNA, known as ROR response elements (ROREs). The ROREs mainly contain the consensus region AGGTCA, and a sequence rich in A/T nucleotides is present immediately before this consensus motif within the regulatory region of target genes on the 5′ end [19]. As a nuclear receptor, RORγ controls gene expression by directly binding to the target genes and also regulates other nuclear receptors and co-regulatory proteins, as well as transcription factors, to act on the target genes [6,7,8,9].

RORγt has been found in IL-17-producing T cells and contributes majorly to the differentiation of the Th17 population, facilitating IL-17A expression through its binding to RORE motifs located within the 2 kb promoter region upstream of the transcription start site, and promotes transcription of IL-17. There is also a conserved non-coding sequence 2 (CNS 2) present about 5 kb upstream of the promoter region of *IL17A*. The CNS2 comprises two ROREs, and both are conserved in humans and mice [42]. It has been reported that RORγt also binds to the CNS2 of *IL17A,* which enables chromatin remodeling to promote transcription. Two other proteins, p300 and JMJD3, bind to the CNS2 (Figure 2) and help maximize the transcription of *IL17A* [43]. RUNX1 also binds to the RORγt and promotes the transcription of *IL17A* (Figure 2). Interestingly, HIF1α not only directly interacts with RORγt but also helps to recruit p300 to promote the *IL17A* expression in Th17 cells [23] (Figure 2). Consequently, RORγt alone is not enough to drive the Th17 differentiation program. RORγt is the master regulator, but other transcription factors are also necessary for the proper establishment of the Th17 program.

## 4. Role of RORγt in Immune Cell Development and Differentiation

### 4.1. Lymphoid Organogenesis

RORγt plays a critical and indispensable role in the development of secondary lymphoid organs, which are crucial for immune function. RORγt is absolutely required for the formation of peripheral and mesenteric lymph nodes, as well as Peyer’s patches [44]. Mice deficient in RORγt entirely lack these organs, underscoring its vital role in their organogenesis [45,46]. Similarly, *RORC* (the gene encoding RORγt)-deficient humans have been observed to lack palpable lymph nodes. RORγt is expressed exclusively in and is essential for the generation and function of fetal lymphoid tissue inducer (LTi) cells during development (Figure 3A) [44,47]. These LTi cells are critical for initiating the formation of lymphoid organs. They activate specialized mesenchymal cells to produce chemokines and upregulate adhesion molecules, which are necessary steps for the proper maturation of lymphoid organs [48]. Retinoic acid also plays a role in inducing RORγt expression, leading to LTi lineage commitment [49].

### 4.2. T Cell Development and Lineage Commitment

RORγt is highly expressed in CD4^+^CD8^+^ double-positive thymocytes [45,50]. This expression is critical for their survival and subsequent differentiation. RORγt promotes the survival of double-positive (DP) thymocytes by upregulating anti-apoptotic proteins, such as Bcl-xL (Figure 3B) [51]. Without RORγt, these thymocytes are more prone to apoptosis, leading to a significant reduction in T cell numbers.

The expression of RORγt in DP thymocytes is transient; its downregulation is essential for their further maturation into single-positive CD4^+^ or CD8^+^ T cells [45,50]. Signals such as those from the IL-7 receptor can inhibit RORγt expression, which is an important step in guiding DP thymocyte differentiation pathways [52].

### 4.3. Lineage Commitment to Th17 Cells

The retinoic acid-related orphan receptor gamma, particularly its isoform RORγt (encoded by the *RORC* gene), is the crucial and indispensable lineage-defining transcription factor for the development and lineage commitment of T helper 17 cells (Figure 3C) [53,54]. Th17 cells are a distinct subset of CD4^+^ T helper lymphocytes known for their production of pro-inflammatory cytokines, including IL-17A, IL-17F, IL-22, and IL-26 [55]. These cytokines play vital roles in host defense against extracellular pathogens but are also significant contributors to the pathogenesis of various autoimmune and inflammatory diseases [56]. RORγt orchestrates the entire differentiation program of these effector cells. Studies have consistently shown that RORγt is both necessary and sufficient for Th17 cell differentiation. T cells lacking RORγt [*Rorc* −/−] completely fail to differentiate into Th17 cells, even when cultured under conditions that normally promote Th17 differentiation. This highlights its absolute requirement for the Th17 lineage. Conversely, forced overexpression of RORγt in naïve CD4^+^ T cells is enough to accelerate the expression of Th17-related cytokines and chemokines, such as IL-17A, IL-17F, IL-22, IL-26, CCR6, and CCL20 [10,57]. The induction of RORγt expression and, consequently, Th17 cell differentiation, is a tightly regulated process influenced by a network of cytokines and transcription factors [58].

IL-6 and TGFβ: These two cytokines are critical for initiating Th17 differentiation. While TGFβ alone induces RORγt, it also induces FOXP3, which can inhibit RORγt function. However, in the presence of IL-6, which activates STAT3 and suppresses FOXP3, the relative level and activity of RORγt are increased, favoring Th17 cell differentiation [53,58].

L-21 and IL-23: IL-21, produced by differentiating Th17 cells, acts in an autocrine manner to upregulate the expression of IL-23R, thereby further stabilizing the Th17 phenotype. IL-23 is not necessary for initial Th17 differentiation but is required to maintain their differentiated state and pathogenicity [58].

STAT3: Activated by IL-6, STAT3 is crucial for Th17 cell differentiation and plays a key role in upregulating RORγt expression. STAT3 activation alone can induce RORγt expression, which is further enhanced when both STAT3 and TGFβ signaling are present [59].

NFAT: NFAT proteins bind to regulatory elements within the *Rorc* gene and activate RORγt transcription in cooperation with NFκB. T cell receptor stimulation can induce modifications at these elements, promoting RORγt expression [60].

RORγt works in concert with other rapidly induced transcription factors, such as basic leucine zipper transcription factor, ATF-like and interferon regulatory factor 4. These factors positively regulate ROR proteins and contribute to Th17 cell specification. The AP-1 transcription factor JunB is also required for Th17 cell differentiation [51,60]. In essence, RORγt serves as the central orchestrator, integrating signals from the cytokine environment and coordinating with other transcription factors to direct the lineage commitment of naïve CD4+ T cells into pro-inflammatory Th17 cells. This pivotal role makes RORγt a key target in understanding and modulating immune responses in various inflammatory conditions.

### 4.4. Influence on Regulatory T Cells Development

While RORγt is most famously known as the master regulator of pro-inflammatory Th17 cells, it also plays a fascinating and complex role in the development of a specific subset of regulatory T cells. These RORγt-expressing Tregs contribute to immune homeostasis, particularly at mucosal sites. Regulatory T cells are primarily defined by the expression of the transcription factor FOXP3, which is essential for their development and suppressive function [61]. However, recent findings have revealed that a distinct subset of FOXP3^+^ Tregs also co-expresses RORγt [62,63]. These FOXP3^+^RORγt^+^ Tregs represent a unique lineage with specific characteristics.

Induced Tregs (iTregs): A considerable fraction of FOXP3^+^ Tregs are induced de novo in the periphery from naive CD4^+^ T cells (pTregs). Under homeostatic conditions, these peripherally induced Tregs are functionally important, especially at feto-maternal interfaces and for maintaining tolerance to food and microbiota-derived antigens at mucosal sites [62]. RORγt^+^ Tregs are particularly prevalent at mucosal surfaces, such as the gut. Here, they play a role in intestinal immune homeostasis [61,63]. Studies suggest that the generation of RORγt^+^ Tregs in the gut can be influenced by specific microbiota [64]. While primarily associated with pTregs, it has been claimed that thymus-derived Tregs (tTregs) can also acquire RORγt expression, especially under pro-inflammatory conditions. A notable feature of RORγt^+^ Tregs is their ability to produce IL-17, although often at lower levels than conventional pro-inflammatory Th17 cells (Figure 3C) [65]. This dual expression of FOXP3 and RORγt, along with IL-17 production, suggests a complex role in immune responses, potentially bridging regulatory and effector functions [57].

FOXP3^+^RORγt^+^ Tregs are characterized by a memory phenotype [CD44^hi^CD62L^lo^] and high expression levels of markers such as ICOS, CTLA-4, IL-10, and IRF4. While their precise suppressive mechanisms are still under investigation, some studies suggest that FOXP3^+^RORγt^+^ Tregs can exert enhanced suppressive capacity during intestinal inflammation compared to FOXP3^+^RORγt^−^ Tregs [61,62]. The development of FOXP3^+^RORγt^+^ Tregs is tightly intertwined with the Th17 differentiation pathway, largely influenced by the cytokine environment.

Balance between TGFβ and Inflammatory Cytokines: In vitro studies show that naïve CD4^+^ T cells, when stimulated under Th17-inducing conditions (which typically include TGFβ and inflammatory cytokines like IL-6, IL-21, and IL-23), can simultaneously express RORγt and FOXP3. The balance between TGFβ and inflammatory cytokines dictates whether cells differentiate into a predominantly Treg or Th17 lineage, or an intermediate FOXP3^+^RORγt^+^ stage [62]. High concentrations of TGFβ tend to favor FOXP3 induction, which can restrain RORγt expression. The stability of the FOXP3^+^RORγt^+^ phenotype can be influenced by factors such as retinoic acid, which contributes to expanding and stabilizing these cells [61].

The role of RORγt in regulatory T cell development highlights the plasticity of T cell lineages. It is central to the generation of a specialized subset of FOXP3^+^ Tregs, particularly at mucosal sites, which can co-express RORγt and IL-17, playing a nuanced role in maintaining immune balance.

### 4.5. RORγt Expression in γσ T Cells

RORγt is expressed in γσ T cells, notably in the IL-17-producing subset, often referred to as γσ T17 cells [66,67]. Unlike some other T cell populations, γσ T cells maintain a high constitutive RORγt expression level [68]. These γσ T17 cells are important producers of IL-17 in various physiological and pathological conditions, including during infections and in the context of tumor microenvironments (Figure 3C) [69].

### 4.6. Development and Lineage Commitment of γσ T17 Cells

Genetic studies in mice have elucidated that γσ T17 cells develop in the embryonic thymus through a stepwise process. This development involves the upregulation of several transcription factors, with RORγt being critical for their lineage commitment, specification, and functional maturation.

Necessity for IL-17 Production: RORγt is directly associated with the production of IL-17A from γσ T cells. Studies using RORγt knockout mice have shown a clear defect in IL-17A production by these cells, underscoring the indispensable role of RORγt in this function [68].

TCR Signaling: T cell receptor signaling is necessary for the development of γσ T17 cells, although research suggests that even weak TCR signals can be sufficient. Interestingly, TCR signaling appears to be more crucial for the transition into an early immature stage of γσ T17 cell development rather than for the initial lineage specification itself [67]. Also, RORγt operates within a complex transcriptional network that dictates γσ T17 cell fate.

STAT Proteins: STAT transcription factors, particularly STAT3, play a role downstream of cytokine and growth factor receptors in regulating γσ T17 cells. STAT3 is critical for the production of IL-17A/F and IL-22 by activated γσ T17 cells [70,71].

SOX13 and cMAF: Along with RORγt, transcription factors like SOX13 and cMAF are upregulated during γσ T17 cell development and contribute to their lineage commitment and maturation [71].

HEB Factors: HEB factors have also been identified as necessary for the specification of fetal IL-17-producing γσ T cells, impacting RORγt expression in these cells [72].

### 4.7. Thymic vs. Peripheral Imprinting

While murine studies indicate thymic development of γσ T17 cells, some research suggests that human γσ T17 cells might acquire their IL-17 bias primarily in the periphery, rather than being fully imprinted during thymic development [73]. This highlights potential species-specific differences in the precise developmental pathways of these cells. RORγt is a central transcription factor for the development of IL-17-producing γσ T cells. It drives their commitment to the γσ T17 lineage, enables their characteristic IL-17 production, and interacts with other signaling pathways and transcription factors to ensure their proper differentiation and function within the immune system.

### 4.8. Role of RORγt in iNKT Cell Development

RORγt plays a crucial and definitive role in the development and functional specification of a particular subset of invariant Natural Killer T (iNKT) cells known as iNKT17 cells. These cells are distinct from other iNKT subsets and are characterized by their capacity to produce interleukin-17 [74].

### 4.9. RORγt as the Lineage-Defining Factor for iNKT17 Cells

Invariant NKT cells differentiate into at least three major effector subsets within the thymus, each distinguished by specific transcription factors and cytokine profiles: iNKT1, iNKT2, and iNKT17 cells [74]. RORγt is the signature transcription factor for iNKT17 cells. Its expression is essential for the development and IL-17 production of this specific iNKT subset. Studies have shown that RORγt is critical for the thymic differentiation of IL-17-producing iNKT cells. Only iNKT cells that express RORγt are capable of massively secreting IL-17. Without RORγt, iNKT cells do not acquire the capacity to produce IL-17 [75]. RORγt acts as a subset-specifying factor for iNKT17 cells. Forced overexpression of RORγt in iNKT cells can redirect their development towards the iNKT17 lineage, demonstrating its sufficiency in driving this fate (Figure 3C) [76].

### 4.10. Thymic Development of iNKT17 Cells

The differentiation of iNKT17 cells occurs primarily in the thymus, following a new thymic pathway that is distinct from those leading to other iNKT subsets. These IL-17-producing iNKT cells are already present in the thymus at an immature stage of iNKT cell development [77].

While iNKT cells share a common developmental precursor with conventional T cells at the double-positive thymocyte stage, RORγt contributes to the survival signals during the alpha chain rearrangement in this DP stage, which is relevant for iNKT cell precursors [77]. The development of RORγt-expressing iNKT17 cells involves the coordinated action of RORγt with other transcriptional regulators.

RUNX1: The transcriptional regulator RUNX1 has been shown to be essential for the generation of RORγt-expressing iNKT17 cells. Its orchestration is crucial for their development in the thymus, spleen, and liver [74].

PLZF: While PLZF (promyelocytic leukemia zinc finger) is a master transcription factor for iNKT cell development and effector functions, iNKT17 cells express intermediate levels of PLZF alongside RORγt [74].

Gene Expression Profile: Single-cell RNA sequencing analyses further confirm that iNKT17 cells exhibit a unique transcriptional profile, expressing high levels of *RORC* and other genes directly dependent on RORγt [78].

Role of RORγt in development and maturation of Group 3 ILCs (ILC3s):

RORγt plays an indispensable role in the development and function of specific subsets of innate lymphoid cells, or ILCs, particularly Group 3 ILCs (ILC3s) that are the key components of the innate immune system, providing rapid responses to various challenges at barrier surfaces [79,80]. ILC3s are a diverse group of cells that includes lymphoid tissue inducer cells (LTi), as well as NCR^+^ and NCR^−^ ILC3s (Natural Cytotoxicity Receptors) (Figure 3A) [81]. RORγt drives the differentiation of ILC3s from their precursor ILC progenitors. Deletion of RORγt leads to a complete absence of ILC3s, highlighting its absolute requirement for this lineage [82]. RORγt^+^ ILC3s are crucial producers of key cytokines such as IL-22, IL-17A, and lymphotoxin-α1β2 (LT α1β2) (Figure 3A). These cytokines are vital for activating epithelial defenses and recruiting polymorphonuclear phagocytes, contributing to immune responses against bacterial infections [83].

Role in Lymphoid Organogenesis: LTi (lymphoid tissue inducer) cells, which are a specialized type of fetal ILC3, are uniquely dependent on RORγt for their development. These cells are essential for the formation of secondary lymphoid organs like lymph nodes and Peyer’s patches during fetal development [84].

Mucosal Homeostasis and Defense: ILC3s are predominantly found at mucosal barrier surfaces, such as in the gut. Here, they play a fundamental role in maintaining tissue homeostasis and orchestrating defense mechanisms against pathogens. Their function in the gut is significantly influenced by the commensal microbiota [85].

In humans, RORγt is expressed in a specific progenitor population found in secondary lymphoid tissues, identified as CD34^+^CD45RA^+^. This RORγt^+^ progenitor is unique in its ability to generate all known human ILC subsets, including natural killer cells [86]. While RORγt can be expressed in other human ILC subsets, its expression is consistently much higher in ILC3s, confirming its primary role in defining this lineage [87]. The roles of ILCs are not rigidly fixed, and they exhibit plasticity, meaning they can interconvert between different subsets depending on the microenvironment.

ILC3 to ILC1 Conversion: RORγt^+^ ILC3s can convert to an ILC1-like phenotype, expressing T-bet and producing IFNγ. This conversion has been observed in vivo and in vitro and can be influenced by specific cytokines like IL-12 and IL-23 [88]. This plasticity provides another layer of immune regulation, allowing ILCs to adapt their function to changing environmental cues [89]. The expression and function of RORγt in ILCs are influenced by various factors (Figure 3).

RUNX3: This transcription factor is required for the expression of RORγt and its downstream target, aryl hydrocarbon receptor, in ILC3 cells [90].

Aryl Hydrocarbon Receptor: AhR, activated by environmental ligands, is another crucial factor that works in concert with RORγt to regulate ILC3 development and function [87,90].

RORγt is an essential transcription factor that dictates the development and function of Group 3 innate lymphoid cells. It enables their characteristic cytokine production and critical roles in lymphoid organogenesis and mucosal immunity and contributes to the dynamic plasticity observed within the ILC family.

## 5. Role of RORγt in Myeloid Cell Development

While the primary role of RORγt is well-established in T cell and ILC development, recent research indicates its involvement in certain aspects of myeloid cell biology, particularly in the context of inflammatory or pathological conditions. Although its involvement in the overall development of myeloid cells, including macrophages, dendritic cells, and neutrophils, is less direct or not yet fully understood, it significantly influences the pathological proliferation and function of specific myeloid populations, such as MDSCs and TAMs, particularly in neoplastic conditions.

Macrophages:

RORγt has been identified to drive “cancer-related myelopoiesis” and “emergency myelopoiesis”. This suggests that under specific conditions, such as in the tumor microenvironment, RORγt plays a role in the expansion and differentiation of myeloid cells. Its ablation in the hematopoietic system has been shown to prevent the generation of tumor-associated macrophages (Figure 3C), which are crucial for tumor progression. This highlights the contribution of RORγt in the development of these immunosuppressive myeloid populations. Furthermore, RORC is upregulated in myeloid cells and is involved in regulating their inflammatory responses, potentially through its role as a receptor for certain metabolic intermediates like desmosterol [91], indicating a regulatory function of RORγt in macrophage activity beyond direct development.

Dendritic Cells:

A subset of RORγt-expressing dendritic cells exists. These RORγt^+^ DCs are considered functionally versatile antigen-presenting cells, capable of sensing pathogens, migrating to lymph nodes, and activating naïve T cells. They can influence T cell tolerance or activation depending on environmental signals (Figure 3C) [92]. However, the current research does not provide extensive detail on RORγt’s specific role in the initial developmental commitment of these DCs from myeloid precursors in a general physiological context.

Neutrophils:

Based on the current search results, there is no direct evidence to suggest a developmental role for RORγt in neutrophils. While other ROR family members, such as RORα, have been implicated in neutrophil migration and activation [93], RORγt’s direct involvement in neutrophil lineage commitment or development was not found.

Myeloid-Derived Suppressor Cells (MDSCs):

Myeloid-derived suppressor cells (MDSCs) are a heterogeneous collection of immature cell types. Murine MDSCs are distinguished by the co-expression of CD11b and Gr-1. Based on the two distinct epitopes of Gr-1, known as Ly6G and Ly6C, MDSCs are classified into two primary subsets based on epitope-specific antibodies: granulocytic MDSCs (CD11b + Ly6GhighLy6Clow) and monocytic MDSCs (CD11b+Ly6ChighLy6Glow). In contrast to murine MDSCs, human MDSCs are distinguished by the expression of CD33 and the low or absent expression of HLA-DR. The prevalent myeloid marker CD11b is also expressed by human MDSCs. The CD11b+ CD33+ HLA-DRlow population can be further categorized into M and G MDSC based on their expression of CD14 and CD15. CD15+ cells exhibit granulocytic morphology, while CD14+ cells display monocytic characteristics [94].

MDSCs were initially identified in the context of malignancy. In healthy individuals, these cells are predominantly found in the bone marrow, though in neoplastic conditions, they readily move to the sites of disease. In cancer, they have an inhibitory effect on several immune system cells, but their impact on T cells has been thoroughly investigated in neoplastic conditions [95]. RORγt is crucial for the generation and proliferation of MDSCs, especially in neoplastic conditions. RORγt drives cancer-related myelopoiesis, which leads to the expansion of tumor-promoting myeloid populations, including MDSCs and tumor-associated macrophages. Studies have shown that ablating *RORC1* in the hematopoietic compartment can prevent the generation of MDSCs and TAMs, thereby inhibiting tumor development (Figure 3C).

RORγt orchestrates the process of myelopoiesis by promoting positive transcriptional regulators of granulopoiesis, such as C/EBPβ, while simultaneously suppressing negative regulators like SOCS3 and BCL3. It also influences key transcriptional mediators involved in the commitment and differentiation of myeloid progenitor cells into the monocytic/macrophage lineage, specifically IRF8 and PU.1 [91]. The expansion of MDSCs in cancer-driven inflammation is enhanced by *RORC* through its collaboration with C/EBPβ, and this effect can occur independently of IL-17A [96].

## 6. RORγt and Th17 Cell Centrality in Chronic Skin Disorders

RORγt is essential for the differentiation and function of Th17 cells, which are a subset of CD4+ T lymphocytes known for producing pro-inflammatory cytokines such as IL-17A, IL-17F, IL-21, IL-22, and GM-CSF. While these cytokines are important for host defense against extracellular pathogens, their dysregulated expression is foundational to the pathology of numerous human inflammatory diseases. RORγt-deficient T cells fail to differentiate into Th17 cells, highlighting their indispensable role [11]. The crucial role of RORγt in driving Th17 cell-mediated inflammation makes it a significant contributor to a wide range of autoimmune conditions like rheumatoid arthritis, multiple sclerosis, systemic lupus erythematosus, and inflammatory bowel disease, as well as chronic inflammatory skin disorders like psoriasis and atopic dermatitis [62].

Psoriasis: Psoriasis is one of the chronic inflammatory skin disorders characterized by excessive proliferation of keratinocytes and a significant infiltration of immune cells. The pathogenesis of psoriasis is intricately linked to the interleukin-23/T helper 17/IL-17 axis, with retinoic acid-related orphan receptor gamma t serving as a pivotal transcription factor in this pathway [97,98,99,100,101,102]. RORγt orchestrates Th17 differentiation and the production of key pro-inflammatory cytokines such as IL-17A, IL-17F, and IL-22 [56]. These cytokines are central to the inflammatory milieu observed in psoriatic lesions [11]. The success of therapies targeting the IL-17 pathway, which inhibit IL-17A and its receptor, further underscores the critical contribution of RORγt-driven Th17 cells to psoriasis development [103,104,105]. Without RORγt, Th17 cells cannot properly differentiate or produce these pathogenic cytokines, highlighting its absolute necessity in the disease process [55].

The cytokines produced by RORγt-dependent Th17 cells directly impact keratinocytes, the predominant cell type in the epidermis. Pro-inflammatory factors released by pathogenic Th17 cells, such as IL-22, TNFα, and granulocyte-macrophage colony-stimulating factor, stimulate keratinocytes to release chemokines. These interactions lead to aberrant keratinocyte activation, hyperproliferation, and tissue inflammation, which are hallmarks of psoriatic plaques [11,56]. This creates a positive feedback loop that sustains the inflammatory cycle in psoriasis beyond Th17 cells.

RORγt expression in other innate immune cells also contributes to the pathogenesis of psoriasis. RORγt-expressing innate lymphocytes, particularly group 3 ILCs, are present in psoriatic skin and can produce IL-17A, IL-17F, and IL-22, contributing to plaque formation. Skin-infiltrating γσ T cells that express RORγt are also significant sources of IL-17A, IL-17F, and IL-22 and have been shown to initiate psoriasiform plaque formation in experimental models [106,107]. A recent study on the murine model of psoriasis revealed that pathogenic MDSCs expressing RORγt produce IL-17, which may also play a role in the development of an inflammatory microenvironment through T cell activation and Th17 differentiation in psoriatic conditions [108]. The collective action of these RORγt-expressing immune cells significantly amplifies the inflammatory response in psoriatic skin, leading to the characteristic clinical manifestations of the disease.

**Atopic dermatitis (AD):** Atopic dermatitis (AD) is a chronic inflammatory skin disorder characterized by eczematous lesions, intense pruritus, and a complex pathophysiology involving immune dysregulation and epidermal barrier dysfunction [109]. While traditionally viewed as a primarily T helper type 2-mediated disease driven by cytokines like IL-4 and IL-13 [110,111], recent research indicates a more intricate immune landscape, with T helper 17 cells and their associated cytokines, particularly IL-17, also contributing to AD pathogenesis [112,113]. The transcription factor retinoic acid-related orphan receptor gamma t plays a crucial role in shaping these Th17 responses.

Although RORγt is required for the differentiation of Th17 cells and their production of pro-inflammatory cytokines such as IL-17A, IL-17F, and IL-22 [106], the role of IL-17 in AD has historically been less understood compared to other Th subsets; growing evidence suggests its involvement in skin inflammation [113]. Studies have reported an increased percentage of Th17 cells in the peripheral blood of AD patients, with this increase correlating with disease severity [112,114]. Similarly, IL-17 is expressed in AD skin lesions, and higher levels of IL-17+ T cells have been found in the dermis of acute AD lesions, indicating its mediating role in AD inflammation [115]. IL-17 coordinates inflammation in local tissues by inducing pro-inflammatory cytokines and chemokines in keratinocytes. This contributes to the recruitment and activation of immune cells, exacerbating the inflammatory cycle in the skin [116].

Beyond its direct inflammatory effects, IL-17 can also contribute to epidermal barrier dysfunction in AD [117]. It has been shown to trigger abnormal keratinocyte proliferation and parakeratosis, processes that impair the skin’s protective barrier. Furthermore, IL-17A, in conjunction with IL-22, may be involved in barrier dysfunction by inhibiting the expression of barrier-related molecules [118].

While RORγt primarily influences Th17 responses, the broader family of ROR nuclear receptors may also play a role in modulating AD. For instance, topical treatment with SR1001, a synthetic RORαγ inverse agonist, has been shown to reduce epidermal and dermal features in mouse models of atopic dermatitis-like disease. This effect was accompanied by a suppression of type 2 cytokines and other inflammatory mediators in lesional skin, suggesting that RORs can influence the overall inflammatory environment in AD beyond just the Th17 axis [119]. Another ROR family member, RORα, has been identified to drive the expression of death receptor 3 in T regulatory cells, which can help restrain allergic skin inflammation [120].

**Systemic sclerosis (SSc)**: Systemic sclerosis (SSc), commonly known as scleroderma, is another complex and severe autoimmune disease characterized by excessive fibrosis of the skin and internal organs, vascular damage, and immune system dysregulation [121,122]. While the precise mechanisms remain under investigation, a growing body of evidence points to a significant role for the transcription factor RORγt and the T helper 17 cell axis in its pathogenesis.

In SSc, there is a consistent report of elevated Th17 cell numbers and increased levels of IL-17 in the peripheral blood, skin, and lungs of patients compared to healthy individuals [123,124]. Studies have specifically shown that both RORγt and IL-17A mRNA expression are significantly higher in the peripheral blood mononuclear cells of SSc patients [123]. This dysregulation of the Th17 pathway is believed to be a key contributor to the extensive fibrosis and persistent inflammation characteristic of SSc [124].

The immune dysregulation in SSc often involves an imbalance between pro-inflammatory Th17 cells and regulatory T cells, which normally suppress immune responses. While Th17 cells express RORγt, Tregs typically express FOXP3. In the presence of pro-inflammatory cytokines like IL-6, IL-1, and IL-21, naive CD4+ T cells are preferentially driven to differentiate into Th17 cells, inhibiting their differentiation into Tregs. Given that IL-6 levels are elevated in SSc patients, this cytokine environment is thought to skew the Th17/Treg balance towards the Th17 subset, thereby exacerbating both fibrosis and inflammation in SSc [125]. The IL-17 produced by RORγt-dependent Th17 cells plays a complex role in SSc, directly influencing the fibrotic process. IL-17 can act as a pro-fibrotic cytokine. It has been shown to induce the expression of transforming growth factor-β and connective tissue growth factor in fibroblasts, both of which are central to collagen production and extracellular matrix deposition, hallmarks of fibrosis in SSc [125]. IFNγ+IL-17+Th17 cells can also promote fibroblast proliferation and enhance collagen-secreting ability through IL-21 production [126]. IL-17 coordinates inflammation in tissues by inducing pro-inflammatory cytokines and chemokines, further contributing to the immune dysregulation seen in SSc [125]. Interestingly, while Th17 cells elicit pro-inflammatory responses, some studies suggest they may also restrain collagen production in certain contexts, indicating a nuanced role that requires further investigation [19]. However, the overwhelming evidence points to their significant contribution to the inflammatory and fibrotic aspects of SSc.

## 7. Targeting RORγt Signaling for Therapy of Chronic Skin Inflammatory Diseases

Chronic inflammatory skin diseases, such as psoriasis and atopic dermatitis, are characterized by persistent inflammation and immune dysregulation. The RORγt has emerged as a crucial therapeutic target due to its central role in driving the T helper 17 cell-mediated immune response, a key pathway in the pathogenesis of these conditions [20,24,127] and offers a promising strategy to modulate the inflammatory cascade. Research is ongoing for the development of novel drugs, chemical inhibitors, and natural products for specifically targeting the RORγt pathway.

### 7.1. Chemical Inhibitors and Novel Drugs

RORγt is considered an attractive drug target for inflammatory diseases, leading to the development of various small molecular modulators, particularly inverse agonists [24,128]. These compounds aim to suppress Th17 differentiation and inhibit the production of pro-inflammatory cytokines like IL-17A and IL-17F, which are significant contributors to skin inflammation and keratinocyte hyperproliferation in conditions like psoriasis [105]. Several RORγt inhibitors have progressed through preclinical and clinical development.

In early studies, oral RORγt inhibitor VTP-43742 has demonstrated a reduction in Psoriasis Area and Severity Index (PASI) scores and a decrease in circulating IL-17A and IL-17F levels, indicating its potential in managing moderate to severe plaque psoriasis in patients (Table 1). A Phase 2a study revealed that the oral application of the drug results in significant elevation of liver transaminases at higher doses (700 mg), which is an indicator of liver damage [97,103,129]. On the other hand, another RORγt inhibitor, JNJ-61803534, has effectively attenuated imiquimod-induced skin inflammation in preclinical models and was also found to be well tolerated at doses up to 200 mg in phase 1 clinical trials, supporting the therapeutic potential of RORγt modulation in skin diseases [103]. An inverse agonist of RORγt, cedirogant, has been developed for the treatment of psoriasis and has undergone Phase I studies to evaluate its pharmacokinetics, safety, and efficacy (Table 1) [104]. Though early Phase 1 studies found cedirogant to be generally well tolerated, with no serious adverse events reported, the efficacy data must be viewed cautiously due to the early discontinuation of the research [130]. A synthetic inverse agonist of both RORα and RORγ, SR1001, when applied topically, demonstrated significant anti-inflammatory potential by reducing epidermal and dermal features of atopic dermatitis-like disease in preclinical mouse models, suppressing the production of type 2 cytokines and other inflammatory mediators in lesional skin without any significant adverse effects (Table 1) [119]. This highlights the potential for broader ROR family modulation in chronic skin conditions. As modulation of RORγt has emerged as a highly promising therapeutic strategy against chronic inflammatory diseases, numerous pharmaceutical companies have focused their efforts on the development of RORγt antagonists and inverse agonists [20]. Some notable oral RORγt inhibitors under investigation include GSK2981278, ABBV-553, ARN-6039, AZD-0284 and JTE-451, suggesting a robust pipeline of potential therapies (Table 1) [92]. These inhibitors work by interfering with co-activator binding to the RORγt ligand-binding domain, thereby impairing its transcriptional activity and reducing the expression of pro-inflammatory Th17 signature genes [21,105].

In spite of promising therapeutic efficacy in targeting RORγt, substantial investment and successful preliminary clinical trials, development of most small-molecule inhibitors has been halted or discontinued, underscoring the myriad of obstacles associated with this strategy [131]. A major concern is the risk of dose-limiting toxicities, resulting in the cessation of numerous prominent medication candidates in clinical trials. A significant worry is that RORγt is crucial for proper T cell development in the thymus. Genetic knockout studies in mice and high-dose animal experiments with early RORγt inhibitors have shown that RORγt disruption can result in thymic atrophy and the emergence of thymic lymphomas [132,133]. Although novel, more selective molecules and other dose regimens are being investigated to reduce this danger, the pleiotropic effects of RORγt on T cell biology continue to pose a significant safety challenge.

Another significant challenge is the specificity of these inhibitors and the possibility of off-target effects. RORγt exhibits significant structural similarities in its ligand-binding region with the analogous nuclear hormone receptor RORα. Numerous early-stage inhibitors demonstrated efficacy against both RORγt and RORα; however, targeting RORα may result in adverse side effects due to its essential role in normal development and metabolic regulation in non-immune tissues such as the brain and liver [22]. Creating small compounds that specifically target the immune-specific RORγt isoform without impacting the widely expressed RORα or other nuclear receptors is challenging.

From a drug discovery standpoint, the characteristics of RORγt ligand-binding pocket also pose distinct challenges, including significant hydrophobicity that favors lipophilic ligands, which frequently have inadequate solubility and metabolic stability [134]. Numerous initial lead compounds faced challenges related to inadequate pharmacokinetics and bioavailability, undermining their efficacy in preclinical investigations and human trials, exemplified by the topical psoriasis candidate GSK2981278 [135]. Recent advancements have concentrated on the formulation of molecules with enhanced qualities; nonetheless, achieving an optimal equilibrium of potency, selectivity, and drug-like attributes continues to pose a significant problem.

### 7.2. Natural Products Modulating RORγt

Beyond synthetic compounds, natural products have also garnered attention for their ability to modulate RORγt signaling with fewer adverse effects, offering potential avenues for novel therapeutic developments.

A preclinical study on an imiquimod-induced murine psoriasis model showed that both oral and topical application of dihydromyricetin, a plant-derived flavonoid, significantly suppresses IL-17 production by pathogenic Th-17 cells and decreases circulating IL-17 levels through direct binding to RORγt protein and reducing its nuclear translocation (Table 2) [136]. Ursolic acid, a natural compound, has been shown to suppress IL17 production by selectively antagonizing the function of the RORγt protein (Table 2) [137]. Correspondingly, a different study reported compounds such as corosolic acid and oleaniolic acid with structural similarities to urosolic acid were able to inversely agonize RORγt through the inhibition of its direct binding to *IL17A* and *IL17F* promoter regions [138]. Digoxin, a cardiac glycoside, has also been identified as an inverse agonist of Th17 cell differentiation, albeit at very high concentrations of 10 μM and 40 μM against mouse Roryt and human RORγT, respectively. However, at nontoxic concentrations of below 100 nM, it was observed that the compound acts as an agonist, increasing binding of RORγt to the promoter regions of both *IL17A* and *IL17F* [139]. In contrast to inhibitors, betulinaldehyde has been identified as a natural RORγt agonist [140]. This highlights the diverse ways natural products can interact with RORs (Table 2).

A variety of other natural substances have also been reported to decrease RORγt mRNA and protein levels or inhibit its transcriptional activity [2]. These findings suggest a rich source of potential lead compounds for further drug discovery efforts. For example, Arctigenin (lignan) can decrease RORγt mRNA and protein levels and has shown amelioration of inflammatory diseases in rodents (Table 2). Epigallocatechin-3-gallate [polyphenol], Astragalus Polysaccharide and Astragaloside IV (saponin), and Oxymatrine (quinolizidine alkaloid) were also observed to have similar effects (Table 2). Macrolide Rapamycin can indirectly affect RORγt by downregulating the transcription factor HIF-1α, which in turn leads to a decrease in RORγt trans-activation (Table 2). α-Mangostin (xanthone) has also been observed to decrease RORγt mRNA and protein levels (Table 2) [2].

## 8. Conclusions

RORγ, especially its isotype RORγt, is regarded as the primary pathogenic factor and the key cellular subset responsible for the production of IL-17A in the development of chronic inflammatory skin disorders. Recent discoveries have altered the prevailing paradigm about the etiology of these inflammatory skin diseases from a Th17-driven condition to an IL-17-driven one. Besides Th17 cells, many immune cells, including natural killer cells, a subset of neutrophils, type 3 innate lymphocytes, and mast cells, are acknowledged to secrete IL-17A, together referred to as “Type 17” cells. The actions of IL-17-producing cells are governed by the master transcription factor, retinoic acid receptor-related orphan receptor gamma t (RORγt). RORγt is regarded as a promising pharmacological target for chronic inflammatory skin disorders, resulting in the creation of numerous small molecular modulators, including inverse agonists. In addition to synthetic drugs, natural materials have also attracted interest for their capacity to influence RORγt signaling, presenting significant opportunities for therapeutic advancement.

However, investigations into RORγt targeting encounter considerable methodological obstacles, especially in the transition from preclinical models to clinical trials in various chronic inflammatory skin disorders. The main challenges arise from the differences in disease pathogenesis and immune mechanisms associated with psoriasis, atopic dermatitis (AD), and systemic sclerosis (SSc). These variations result in inconsistencies in research models, target engagement evaluation, and overall therapeutic results. Preclinical models frequently do not accurately reflect the complexity of human skin diseases, as notable discrepancies between mouse models and human clinical trials pose a considerable challenge.

While animal models, including the imiquimod-induced dermatitis model for psoriasis, effectively replicate essential pathological characteristics, notably the Th17-mediated inflammation driven by IL-17 and IL-23, other conditions, such as AD, are influenced by a complex interaction of immune pathways, primarily Th2 and Th22, in conjunction with Th17. In such instances, focusing solely on RORγt may prove inadequate, requiring alternative methodological strategies that consider the wider context of immune dysregulation. Although certain humanized mouse models are available, numerous mouse models do not adequately represent the complete range of human diseases, particularly regarding T cell subsets, cytokine profiles, and interactions with the skin barrier. This may result in inaccurate efficacy data, as a drug that appears effective in a simplified animal model may not succeed in complex human trials. Evaluating the efficacy of an RORγt inhibitor in modulating the Th17 pathway across various skin diseases necessitates specific methodological approaches and the identification of relevant biomarkers. In psoriasis, the prominent Th17 component establishes IL-17 and IL-22 as reliable biomarkers for target engagement. Research indicates a decrease in these cytokines within psoriatic plaques subsequent to treatment. The efficacy of IL-17-blocking antibodies establishes a definitive standard for assessing this methodology. Conversely, due to the mixed Th2/Th17 characteristics of AD, evaluating RORγt inhibition presents greater complexity. Although a reduction in IL-17 is an important indicator, it is essential for researchers to also assess additional cytokines such as IL-4, IL-13, and IL-31 to comprehensively evaluate the drug’s effect on the immune response. This necessitates a more thorough, multi-pathway analysis in clinical studies.

The diverse methodological approaches lead to inconsistent clinical outcomes among various skin conditions. Psoriasis exhibits a greater reliance on the Th17 axis compared to atopic dermatitis (AD) or systemic sclerosis (SSc). Therefore, RORγt inhibitors that are ineffective in treating psoriasis may still be viable for other conditions, provided that their off-target effects are manageable or their mechanisms can be integrated with alternative therapies. The inhibition of RORγt is subject to considerable variation depending on the method of delivery employed. Topical application is frequently favored for skin diseases to minimize systemic exposure and reduce the risk of toxicity. Developing effective topical formulations, exemplified by the failure of different RORγt inhibitors, involves specific methodological challenges concerning drug penetration and stability. The future of RORγt inhibition may depend on improved patient stratification utilizing biomarkers, an area that is currently under investigation and developing methodologies for identifying patients with diseases significantly influenced by Th17, including specific forms of chronic inflammatory skin disorders like psoriasis, atopic dermatitis and systemic sclerosis. Studies focusing on RORγt should also consider the intricate biology of the Th17 pathway, as it constitutes a dynamic target. Although RORγt serves as a principal regulator, it operates in conjunction with numerous other transcription factors. Additional cell types, aside from Th17 cells, also express RORγt and play a role in the immune response, including type 3 innate lymphoid cells (ILC3s) and γδ T cells. Merely inhibiting RORγt may not adequately resolve the complete pathology of specific diseases; in certain circumstances, such as inflammatory bowel disease, extensive IL-17 suppression has demonstrated diminished efficacy or even adverse effects. Moreover, RORγt has a role in safeguarding Th17 responses against specific pathogens, resulting in a possible trade-off between managing autoimmune diseases and inhibiting immune function. Consequently, future research should be focused on investigating more targeted approaches, including the development of dual-targeting medicines or inhibitors that regulate certain RORγt-dependent pathways without inducing adverse side effects.

## Figures and Tables

**Figure 1 biomolecules-15-01543-f001:**
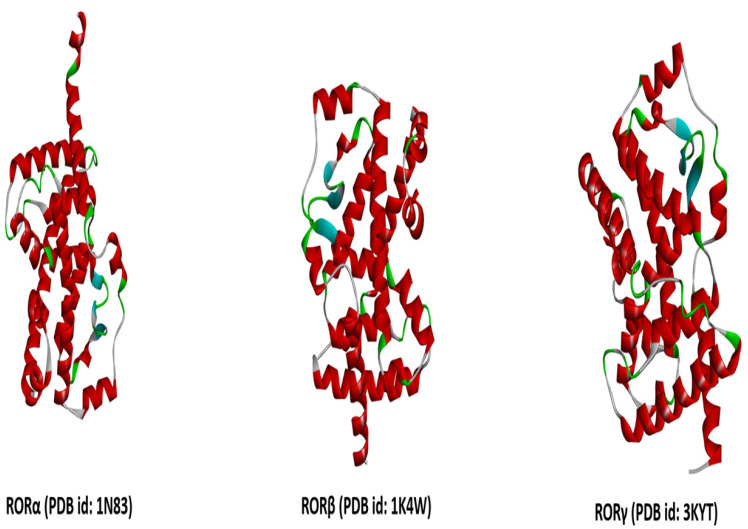
Crystallographic structure of human RORα, RORβ and RORγ proteins. PDB files were downloaded from the RCSB-PDB database “https://www.rcsb.org/ (accessed on 12 September 2025)” and prepared for visualization (attached ligands and water molecules removed) in the Biovia Discovery Studio visualizer tool “https://discover.3ds.com/discovery-studio-visualizer-download (accessed on 12 September 2025)”.

**Figure 2 biomolecules-15-01543-f002:**
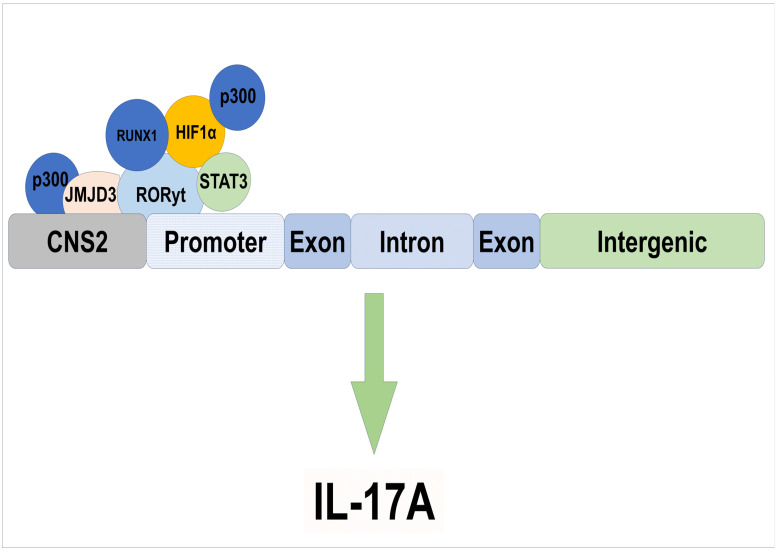
An overview of transcription factors regulating the expression of IL-17 production. Transcription factors like STAT3, p300, and JMJD3 directly or indirectly bind to specific loci and help to promote IL-17A production, as RORγt alone is not enough to drive differentiation of Th17 cells.

**Figure 3 biomolecules-15-01543-f003:**
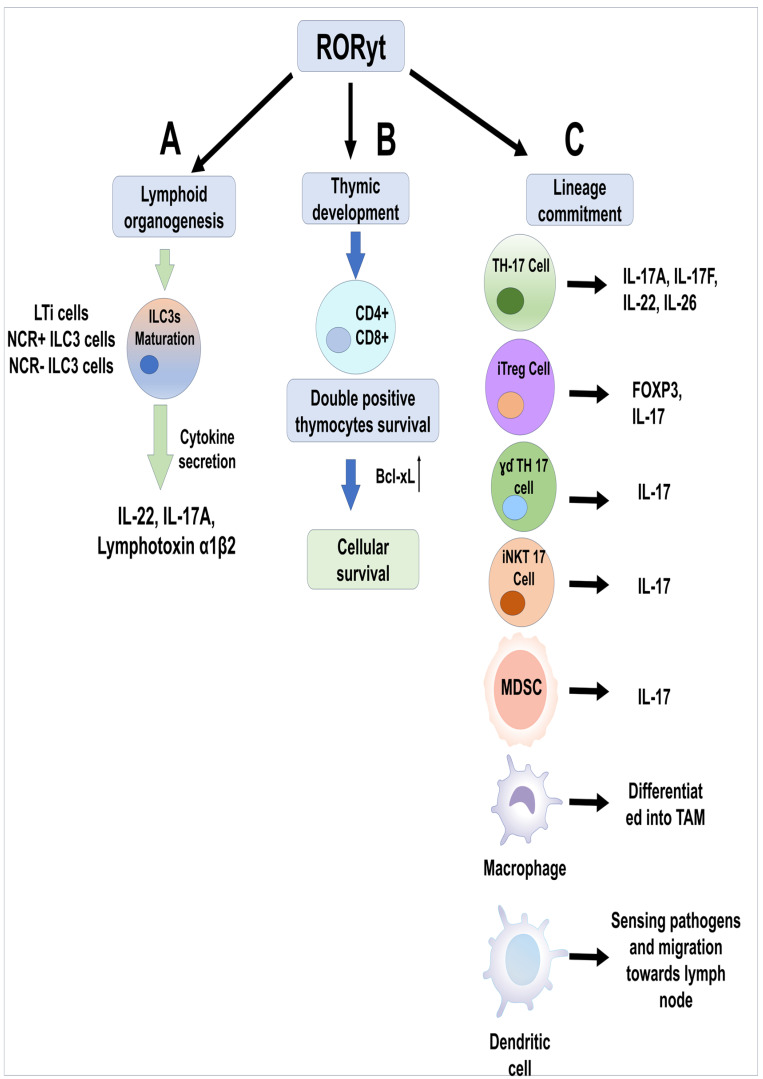
(**A**) RORγt is indispensable for the maturation of LTi (lymphoid tissue inducer) cells that produce IL-17A, IL-22 and lymphotoxin-α1β2, which are vital for activating epithelial defenses and recruiting polymorphonuclear phagocytes. (**B**) RORγt also regulates the expression of anti-apoptotic proteins like Bcl-xL, which is essential for double-positive T cell survival at the thymus. (**C**) RORγt not only plays a central role in the differentiation of just Th17 lymphocytes but is essential for different innate and adaptive immune cell development and function.

**Table 1 biomolecules-15-01543-t001:** Novel drugs and small molecular inhibitors that directly target RORγt signaling.

SL No.	Chemical Inhibitors and Novel Drugs	Mode of Action	Trial Phase	References
1.	VTP-43742	Selectively inhibits RORα and RORβ isotypes	Phase 2a	[97,103]
2.	JNJ-61803534	Selectively inhibits RORγt	Phase 1	[103]
3.	Cedirogant	Inverse agonist of nuclear receptor ROR-gamma isoform 2 (RORyt)	Phase 1	[104]
4.	SR1001	Inverse agonist of RORα and RORγ	Preclinical	[119]
5.	GSK2981278	Interferes with co-activator binding to the RORγt ligand-binding domain	Phase 1/Phase 2	[97]
6.	ABBV-553	Interferes with co-activator binding to the RORγt ligand-binding domain	Phase 1/Phase 2	[97]
7.	ARN-6039	Interferes with co-activator binding to the RORγt ligand-binding domain	Phase 1	[97]
8.	AZD-0284	Interferes with co-activator binding to the RORγt ligand-binding domain	Phase 1	[97]
9.	JTE-451	Interferes with co-activator binding to the RORγt ligand-binding domain	Phase 1/Phase 2	[97]

**Table 2 biomolecules-15-01543-t002:** Natural substances that directly target RORγt signaling.

SL No.	Natural Products	Mode of Action	References
1.	Dihydromyricetin	Reduces nuclear translocation of RORγt	[136]
2.	Ursolic Acid	Selective antagonist of the RORγt	[137]
3.	Corosolic acid	Inverse agonist of the RORγt	[138]
4.	Oleaniolic acid	Inverse agonist of the RORγt	[138]
5.	Digoxin	Suppresses Th17 cell differentiation by inversely agonizing RORγt activity at very high doses	[2]
6.	Arctigenin [lignan]	Decreases RORγt mRNA and protein levels	[2]
7.	Epigallocatechin-3-gallate [polyphenol]	Decreases RORγt mRNA and protein levels	[2]
8.	Astragalus Polysaccharide and Astragaloside IV [saponin]	Decreases RORγt mRNA and protein levels	[2]
9.	Oxymatrine [quinolizidine alkaloid]	Decreases RORγt mRNA and protein levels	[2]
10.	Rapamycin [macrolide]	Indirectly affects RORγt by downregulating the transcription factor HIF-1α	[2]
11.	α-Mangostin [xanthone]	Decreases RORγt mRNA and protein levels	[2]

## Data Availability

No new data were created or analyzed in this study.

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
