# Peer review of "Critical Evaluation of the Role of Transcription Factor RAR-Orphan Receptor-γt in the Development of Chronic Inflammatory Dermatological Diseases: A Promising Therapeutic Target"

_biomolecules, 2025, doi:10.3390/biom15111543_

Round 1

Reviewer 1 Report

Comments and Suggestions for Authors

The review by Pramanik focuses on RORγt’s regulation of IL-17–driven inflammation and explores targeting the RORγt–IL-17 axis as a potential therapy for chronic skin inflammation. It is known that elevated IL-17 levels are linked to chronic inflammatory skin diseases by promoting immune cell recruitment and tissue damage. Although the topic is interesting, the manuscript is poorly written. It contains numerous errors, uses incorrect nomenclature, and fails to cite several important articles. Therefore, it requires substantial revision.

  1. Introduction section (Page 1, lines 39–42): The authors should include more appropriate references, such as DOI: 10.1016/j.bbcan.2023.189021 and DOI: 10.3390/ijms21155329.
  2. Line 46: RUNX1 and STAT3 are not co-regulatory proteins; they are transcription factors. RORγ interacts with SRC-1, SRC-2, SRC-3, p300, and the transcriptional co-activator with PDZ-binding motif (TAZ) (doi: 10.4049/jimmunol.1600548; doi: 10.1038/nature10075; doi: 10.4049/jimmunol.1801187; doi: 10.1016/j.jmb.2021.167258; doi: 10.1038/srep16355; doi: 10.1016/j.imlet.2018.07.004; doi: 10.1038/ni.3748).
  3. Nomenclature: The authors use incorrect and inconsistent nomenclature.
  • Use IL-6, IL-17A, and IL-17F when referring to cytokines.
  • Use IL17A and IL17F when referring to genes.
  • Use FOXP3 and RUNX1 for human proteins.
  • RORC denotes the human gene, whereas Rorc refers to the murine gene.
  1. Page 2, lines 58–59: Include appropriate references.
  2. Page 2, lines 61–62: Include appropriate references.
  3. Page 2, lines 82–83: The statement “only RORγ is predominantly expressed in immune cells, particularly in the thymus” is incorrect. RORγt is the isoform predominantly expressed in immune cells.
  4. Lines 91–93: Include the following references: DOI: 10.1038/s41467-019-12529-3; DOI: 10.1371/journal.pgen.1004331; DOI: 10.1152/physiolgenomics.00098.2007; DOI: 10.1621/nrs.07003; DOI: 10.1093/nar/gks630.
  5. Lines 108–116: This section lacks proper citations. For example, see DOI: 10.1124/pharmrev.121.000436.
  6. Lines 128–129: Gene names should be italicized.
  7. Figure 1: The upper panel should be removed; it is incorrect and misleading as it does not correspond to specific transcripts.
  8. Lines 155–156: There is repetition, and RUNX1 and STAT are incorrectly listed as co-activators. Please remove this.
  9. Line 209: Nomenclature is incorrect again — please revise.
  10. Lines 213–215: Include appropriate references.
  11. Line 255: Clarify how the murine protein FoxP3 could be co-expressed with human RORγt.
  12. Lines 363–365, 413–414: The phrase “ROR is crucial” is repeated numerous times without context. Please specify what ROR is crucial for.
  13. Lines 420–429: The authors fail to distinguish which RORγ isoform is relevant in macrophages.
  14. Line 461: Ensure consistent use of terminology: RORγ, RORgamma, or RORC1. Remember that RORC is a gene—it cannot “drive” a process; only its protein product can.
  15. Lines 485–491: Why are the original papers demonstrating the link between Th17 cells and psoriasis not cited?
  16. Lines 491–492, 565–567: Repetitive statements again emphasize that RORγ is crucial. Please consolidate and avoid redundancy.
  17. Section 7: This section is too superficial. The authors should discuss potential side effects of the compounds used, their efficacy or lack thereof, and other relevant pharmacological considerations.
  18. Line 618: “RORgammadelta”? This is incorrect—please verify and correct.
  19. Section “Natural products modulating RORγt”: The discussion lacks critical analysis and omits several key references. For example, when discussing ursolic acid, the authors should also mention oleanolic and corosolic acids (DOI: 10.3390/ijms23031906). Moreover, this paper demonstrates that these compounds could have ectopic applications in several inflammatory conditions.
  20. Digoxin: This section demonstrates a lack of understanding. Digoxin was identified as an inverse agonist at a concentration of 40 µM, which is toxic for humans and animals. A Polish group showed that at concentrations up to 100 nM, it acts as an agonist (DOI: 10.3389/fphar.2018.01460). Similar effects have been observed for other cardiac glycosides.
  21. Table 2: Digoxin should not be classified as an antagonist.
  22. Line 657: RORC cannot be transactivated—it is a gene.
  23. Conclusions section: The conclusion is too superficial. The authors should discuss the current understanding of RORγt inhibition, summarize clinical trial outcomes, and outline future perspectives and directions for research.
  24. Language and Grammar: The manuscript contains numerous grammatical errors and awkward phrasing. A thorough revision by a native English speaker or professional editing service is strongly recommended.
    Any papers recommended in the report are for reference only. They are not mandatory. You may cite and reference other papers related to this topic.
Comments on the Quality of English Language

The manuscript should be corrected by a native speaker.

Reviewer 2 Report

Comments and Suggestions for Authors

The authors analyzed a large number of literature sources, summarized the information, and demonstrated the role of RORγt immune cell development and differentiation, as well as the involvement of this transcription factor in the pathogenesis of chronic inflammatory skin diseases.

The reviewer has a number of minor comments:

  1. Title: If this is a critical evaluation, then the problems and challenges in therapy targeting RORÉ£t should be added. Issues regarding differences in methodological approaches to research, if any, can also be added.
  2. Conclusion: I would like to see in the conclusion or in a separate section the similarities and differences in the involvement of RORγt in the pathogenesis of various inflammatory skin diseases. I would also like to highlight the existing specific gaps in research related to the three skin diseases mentioned.
  3. Remove the abbreviation in parentheses from the title.
  4. Judging by the abstract and introduction, the article primarily concerns RORɣt. In this case, the ɣ in the manuscript title should be corrected to ɣt.
  5. Page 1, line 18, insert "The expression of STAT3."
  6. Lines 24 and 25: Correct RORÉ£T to RORÉ£t.
  7. Page 2: p. 47 – abbreviate "The Retinoic acid-related Orphan Receptor gamma."

Remove the expanded version from the text, as the abbreviation has already been introduced. This also applies to other abbreviations.

  1. Line 77 – replace gamma with the symbol. "Isoforms, upstream ligands, and ROR gamma activation" - remove "and ROR gamma"; remove the colon at the end here and in all headings.
  2. Lines 85 and 86: where the gene is mentioned, change ROR C to italicized lowercase. Do this throughout the text of the article. Also check the writing format of all gene designations.
  3. Line 116 - post-translational with a hyphen
  4. Line 128 - gene names in italics
  5. Figure 1 caption - indicate the source of the data for the figures or the figure itself
  6. Line 142 - replace gamma with the symbol
  7. Line 155 - co-regulatory with a hyphen
  8. Figure 2 looks a bit chaotic. Organize it as much as possible. Replace T with t in the name of the RORÉ£t factor.
  9. Line 175 - remove Colon
  10. Lines 212–213 – This phrase is repeated many times in different parts of the article (lines 212, 491, 534, 565). Remove repetitions.
  11. Lines 176–179 – Too general a phrase. Remove or clarify with regard to RORγt.
  12. Line 235 – Gene name – in lowercase and italics. Remove explanation in parentheses.
  13. Replace all parentheses with parentheses (except for references).
  14. Lines 249–255 – Treg – replace subscript with lowercase.
  15. Lines 328 and 333 – Number subheadings.
  16. Lines 445–462 – If the abbreviation "MDSCs" is used, do not write "Myeloid-derived suppressor cells" in full multiple times.
  17. Line 489 - RORγt's - remove 's
  18. In the figures, replace the font with a sans-serif font (Arial or similar).
  19. Figure 3 - Replace T with t in the factor name RORÉ£t.
  20. Line 500 - typo in the word "components"

Round 2

Reviewer 1 Report

Comments and Suggestions for Authors

The authors substatially revised their manuscipt, thus I have no further comments.